# QUANTIZATION MEETS SPARSIFICATION FOR FASTER IMAGE GENERATION

## ABSTRACT

With the advancement of deep learning, various generative models have emerged in recent years. Diffusion models, notably exemplified by the Diffusion Transformer (DiT), have become a key component in generative modeling, demonstrating outstanding performance in vision generation tasks. In inference scenarios for generative tasks, quantization and sparsification are widely used techniques to reduce the memory consumption and computation cost. However, existing methods focus on only one of these techniques, and it remains underexplored whether we can leverage the strengths of both.

To fill this gap, this work develops a brand new acceleration framework that applies offline sparsification and quantization to DiT models, facilitating faster image generation while preserving generation quality. Furthermore, we develop a novel and efficient matrix multiplication kernel that leverages low-bit and sparse computing capabilities of Tensor Cores. We conduct experiments on both the 12B open-source FLUX.1-dev model and the 18B closed-source MoE model from our industrial partner. Empirical results show that our kernel achieves a speedup of 1.64-2.16×, delivering an efficiency improvement of 1.09-1.35× to the end-to-end workflow, while incurring negligible degradation in generation quality.

## 1 INTRODUCTION

Generative tasks represent a major category within deep learning. With the advancement of deep learning, especially the development of Transformers (Vaswani et al., 2017) and diffusion models (Ho et al., 2020), deep learning models based on the Diffusion Transformer (DiT) (Peebles & Xie, 2023) architecture have achieved increasingly impressive performance on vision generation tasks.

Alongside the development of generative models, model sizes have continued to grow. Early models like SD 1.5 (Rombach et al., 2022) had only 0.86 billion parameters, while the FLUX.1-dev model (Labs, 2024) has reached 12 billion parameters, and future models are expected to grow even larger (Team, 2025; Wu et al., 2025; Gao et al., 2025). While larger models bring improvements in generation quality and instruction following, they also incur higher deployment and inference costs. Therefore, reducing the cost of generative tasks has become a critical research direction—specifically, how to effectively compress models to reduce memory footprint and improve inference efficiency while preserving generation quality.

Quantization and sparsification are two widely used model compression techniques. Typically, quantization (Jacob et al., 2018; Nagel et al., 2019; 2020; Lin et al., 2020) reduces model size by storing parameters in lower-precision data types, while sparsification (LeCun et al., 1989; Han et al., 2015; Frantar & Alistarh, 2023; Fang et al., 2025; Xia et al., 2025) decreases the total number of parameters by removing less important weights. Deploying these compression methods typically requires corresponding system-level support: quantization demands kernels for quantization, dequantization, and low-precision matrix multiplication, while sparsification requires kernels that support sparse matrix operations. Nevertheless, existing compression approaches generally focus on only one of these techniques and rarely combine both. NVIDIA has proposed a method that combines quantization with sparsification by introducing sparse Tensor Cores (Mishra et al., 2021), but it is designed for traditional CNN models and based on Per-Tensor quantization, which performs poorly on large scale diffusion models.

To fully leverage the benefits of both quantization and sparsification while preserving generation quality, this paper presents a novel inference acceleration framework based on finer-grained Per-Token quantization and sparsification. Our main contributions are as follows:

- We apply offline pruning and quantization to multiple models, and build a complete workflow for model quantization and sparsification, enabling rapid deployment of compression on arbitrary DiT models through module replacement.
- We design and implement an efficient quantized sparse GEMM kernel, namely QuaSpa, that supports per-token quantization and exploits the sparse computing capabilities of Tensor Cores.
- We experimentally demonstrate the efficiency of our kernel, and our method can effectively accelerate generation efficiency while maintaining image generation quality.

## 2 RELATED WORKS

### 2.1 DIFFUSION MODEL

Diffusion models are a class of AI techniques based on probabilistic generative models, mainly used for high-quality image generation, audio synthesis, and data augmentation. The core idea of the diffusion model involves learning the data distribution through a forward process of gradually adding noise and a reverse process of denoising, enabling the synthesis of new samples.

In recent years, diffusion models have achieved remarkable success in the field of generative AI, gradually replacing traditional Generative Adversarial Networks (GAN) (Ian J. Goodfellow, 2014) and becoming one of the dominant approaches in AI-generated content (AIGC).

Existing diffusion models are built upon various architectures, among which two types constitute the majority: one represented by SD1.5 (Rombach et al., 2022) and SDXL (Podell et al., 2024) based on the U-Net (Ronneberger et al., 2015) architecture, and the other represented by SD3 (Esser et al., 2024) and FLUX.1-dev (Labs, 2024) based on the Diffusion Transformer (DiT) architecture.

### 2.2 DIFFUSION TRANSFORMER

Diffusion Transformer (DiT) is a diffusion model based on the Transformer architecture, designed for various image and video generation tasks. DiT demonstrates the effectiveness of integrating the Transformer paradigm with diffusion models and further validates the strong scaling capability of the Transformer architecture within this context. As the DiT model size is systematically increased and the quality of training data improved, its generative performance consistently improves.

Multimodal Diffusion Transformer (MMDiT) (Esser et al., 2024) is a model architecture developed from the DiT framework. Traditional text-to-image diffusion models typically inject text embeddings into the image Transformer via cross-attention mechanisms, while sharing the same set of weights between modalities, which limits the full potential of each modality. MMDiT addresses this by assigning separate Transformer weights for text and image modalities, enabling independent computation in the MLP layers. However, in the attention layers, the activations from both modalities are concatenated for joint processing. This architecture enables a more effective fusion of multimodal information, leading to improved instruction following and overall generation quality.

### 2.3 MODEL QUANTIZATION

Model quantization refers to the process of converting the weights, activations, and other parameters of a deep learning model that originally stored and computed in high-precision data types (e.g., FP32, FP16) into low-precision data types (e.g., FP8, INT4). This reduces the size of model, lowers computational cost, and accelerates inference. In general, quantization represents a high-precision weight as a combination of a low-precision quantized weight and shared scale factors, reconstructing the original weight through the product of the quantized value and its corresponding scale.

Works such as GPTQ (Frantar et al., 2023) and AWQ (Lin et al., 2024) have been devoted to model quantization, and we can classify these quantization methods according to the granularity and precision of quantization.

Based on granularity, existing quantization strategies are typically classified as per-tensor, per-token, or per-block quantization. Per-tensor quantization (Sakr & Shanbhag, 2019; Nagel et al., 2019) applies a single scale factor to an entire tensor. Per-token quantization (Wei et al., 2022; Xiao et al., 2023) computes a separate scale factor for the weights associated with each token. Per-block quantization (Dettmers et al., 2022b; Jang & Tambe, 2025) divides the tensor along the dimension of token into multiple blocks, computing a unified scale factor for all tokens within each block, which offers finer granularity compared to per-token quantization. Figure 1 illustrates the differences among the three quantization methods. For generative models, finer-grained quantization strategies generally yield better performance.

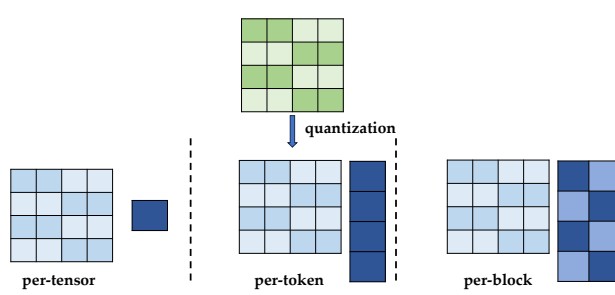

Figure 1: Different types of quantization.

Quantization precision is another critical metric in quantization methods. It involves two aspects: the data type used for model weight storage and the data type used for activations during computation. The model weight storage type (denoted as W) refers to the data type of the quantized model weights, while the activation type (denoted as A) refers to the data type used during computation. For example, W4A16 (Lin et al., 2024; Zhao et al., 2024) indicates that model parameters are stored in 4-bit formats (e.g., FP4, INT4) while computations are performed in 16-bit formats (FP16 or BF16). Since storage and computation data types differ, an additional dequantization step is required before computation and we generally refer to this type of method as weight-only quantization. In contrast, W8A8 (Dettmers et al., 2022a; Kuzmin et al., 2022; Shen et al., 2024) denotes that both storage and computation use 8-bit formats (e.g., FP8, INT8), eliminating the need for dequantization before computation, which is shown in Figure 2a. This type of method is generally known as activation-weight quantization. In practice, using low-precision storage primarily reduces the memory footprint and memory access latency, while low-precision activations improve computational efficiency.

## 2.4 SPARSE GEMM

Sparse general matrix multiplication (GEMM) refers to a specialized form of matrix multiplication where one or both input matrices contain a significant number of zero elements. During computation, additional metadata, such as the positions of non-zero elements, is typically required to represent these matrices efficiently. Numerous studies have explored effective representations for sparse matrices (Cohen, 1998; Buluç & Gilbert, 2008; Demirci & Aykanat, 2020; Zhou et al., 2021). However, due to the strong dependence of matrix multiplication efficiency on hardware, general-purpose sparse matrix multiplication implementations often achieve suboptimal computational performance.

Zhu et al. (2019) proposed a scheme to accelerate sparse matrix multiplication using Tensor Cores. Starting with the NVIDIA Ampere architecture (NVIDIA, 2020), structured sparsity was introduced, offering enhanced suitability for deep learning inference acceleration. This capability is primarily delivered through Tensor Cores in NVIDIA GPUs (Mishra et al., 2021), supporting a 2:4 sparsity pattern—where, in every group of four adjacent weights, at least two are zero, achieving a 50% sparsity rate. The structured pattern (Teng & Wang, 2022) enables efficient memory access, effective model inference acceleration, and relatively easy recovery of model accuracy. Conventional computation methods are often inefficient when processing such structured data, whereas specialized optimizations for sparse matrix operations enable highly efficient processing, significantly improving computational efficiency and reducing energy consumption. To utilize sparse Tensor Cores, we require preprocessing of the sparsified parameters and the workflow is illustrated in Figure 2b.

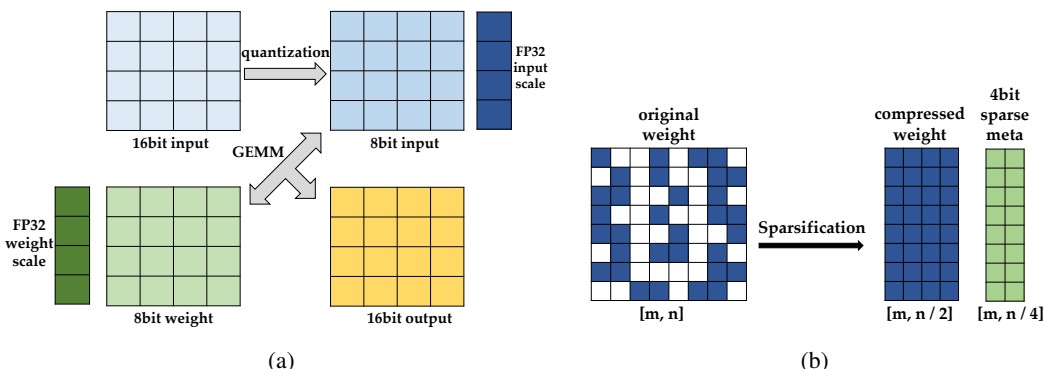

Figure 2: (a): Example of per-token quantization W8A8 GEMM. (b): Preprocessing workflow of the sparsified parameters.

# 3 METHOD

## 3.1 PRUNING FOR SPARSE MASK

As mentioned in Section 2.4, to leverage the computational capacity of sparse Tensor Cores, we need to provide a 0-1 sparse mask for each weight that is to be sparsified. Since the sparse Tensor Core only supports the 2:4 sparsity pattern, where exactly two out of every four adjacent elements are ones and the other two are zeros, the provided sparse mask must conform to this format.

The sparse mask can be obtained in a manner analogous to model pruning, by setting relatively less important weights to zero. We follow existing offline model pruning methodologies (Li et al., 2023; Sun et al., 2024; Yang et al., 2025), acquiring the sparse masks based on these approaches without modifying the original model weights. Once the sparse masks for all model parameters are obtained, we can leverage sparse computation for efficient model inference.

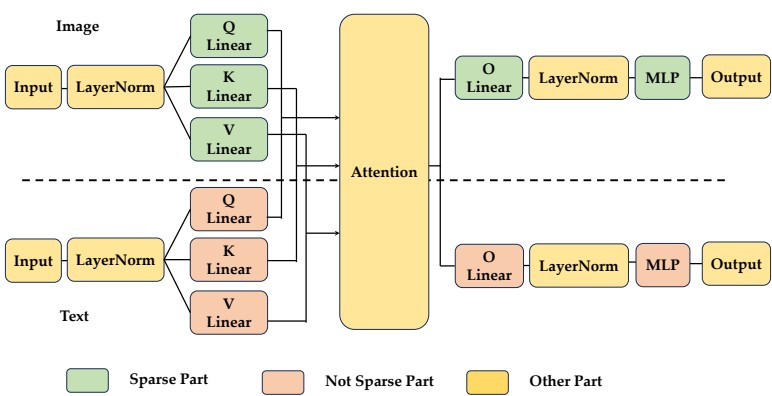

Figure 3: The structure of a MMDiT block and our sparsification strategy on a MMDiT block.

An MMDiT block includes two modalities: text and image. Each modality has its own matrix multiplication weights. We observed in practice that the text modality accounts for a relatively small portion of the computation. Moreover, sparsifying the weights in the text modality significantly impacts the generation quality. Therefore, in most cases, we only apply quantization to the matrix multiplication weights in the image modality. In consideration of generation throughput and quality, our sparsification strategy is illustrated in Figure 3.

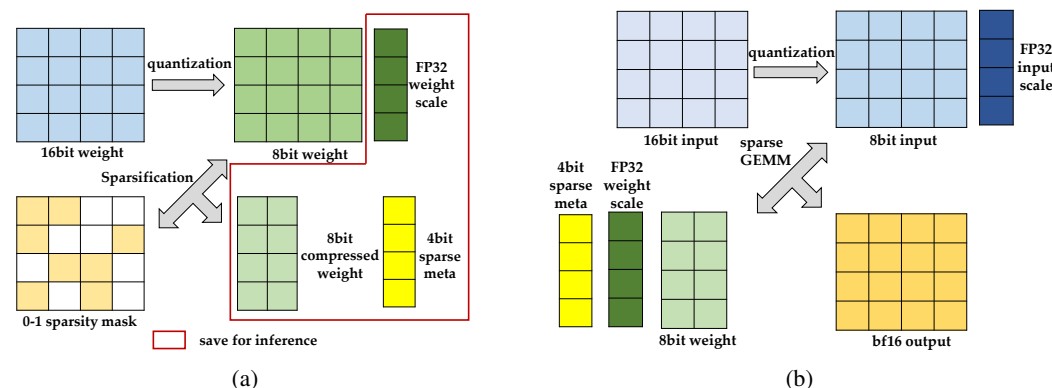

Figure 4: (a): Preprocessing workflow of quantization and sparsity. (b): Our W8A8 sparse GEMM inference workflow.

## 3.2 Dynamic Quantization and Sparsity in Inference

At the infrastructure level, we have implemented a workflow that adaptively deploys quantization and sparsification inference pipelines based on the model, enabling acceleration of diverse DiT models through sparsification and quantization. This section illustrates our workflow for combining quantization and sparsity during model inference.

To minimize the impact of quantization and sparsification on output image quality, our pipeline adopts a deployment strategy that applies quantization first, followed by sparsification. As illustrated, the model parameter initialization process consists of the following three steps. We employ a layer-wise parameter loading workflow to minimize peak GPU memory consumption:

1. Load the original model weights (in FP16 / BF16).

2. Quantize the weights to the target precision (FP8). For a weight tensor $w$, this yields the quantized weights $w_q$ and the corresponding scale factors $w_s$.

3. Apply sparsification based on the sparse mask associated with each parameter. Positions where the mask value is 1 are retained, while positions with a mask value of 0 are discarded. Since our masks conform to the 2:4 sparsity pattern, a parameter tensor of shape $[m, n]$ is transformed into shape $[m, n/2]$ after sparsification, reducing its memory footprint by half. Additionally, we compute the sparse metadata required by the sparse matrix multiplication kernel based on the sparse mask, which is then passed as input to the kernel.

Our deployment workflow is shown in Figure 4a.

Throughout the inference pipeline, we adopt a dynamic quantization scheme, which quantizes high-precision activations $x$ to low-precision $x_q$ and computes the corresponding scale factor $x_s$ on-the-fly before each GEMM operation. Our sparse matrix multiplication kernel takes the inputs $x$, $w$, meta, $x_s$, and $w_s$, and computes the output $o$. The data types and shapes of each component are illustrated in the figure.

Since both the sparse GEMM and dense GEMM kernels have the same input format requirements for activations, no additional modifications are needed in this part. The workflow of our FP8 sparse GEMM is shown in Figure 4b.

## 3.3 Sparse Gemm Kernel with Per-Token Quantization

After introducing both quantization and sparsification, the overall GEMM workflow also changes. Compared to the original GEMM, the new workflow introduces additional computational overhead:

(1) Per-token quantization introduces extra scale factors: for the output matrix of the GEMM kernel, we must multiply by the scale factors of both the input and the weight to obtain an accurate output matrix.

(2) GEMM follows the rule that a row-wise input matrix A multiplied by a column-wise input matrix B produces a row-wise output matrix D. In model inference, matrix A typically represents activations (input) while matrix B corresponds to model weights (weight). Sparsification requires the use of sparse Tensor Core Matrix Multiply-Accumulate (MMA) instructions. However, the sparse Tensor Core only supports sparsification on matrix A. Therefore, we must swap the positions of the input matrices, treating the model weights as matrix A and the activations as matrix B. After this swap, to obtain the correct GEMM result for subsequent computations, the resulting matrix D must be transposed.

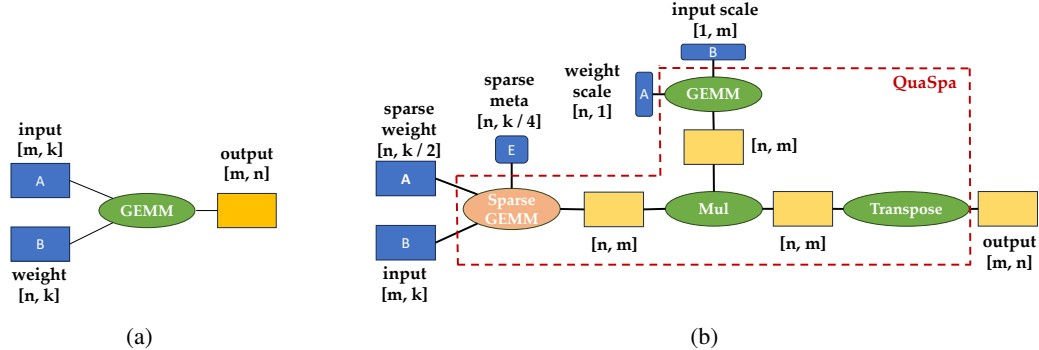

(a)                     (b)

Figure 5: (a): Workflow of original GEMM kernel. (b): Workflow of sparse GEMM kernel with per-token quantization.

Our workflow is illustrated in Figure 5b. Compared to the original GEMM in Figure 5a, we introduce three additional kernels. If the kernels are executed independently, they will incur extra memory access and kernel launch overhead. To maximize inference efficiency, we fused these kernels together and designed a hardware-friendly sparse GEMM kernel with per-token quantization, which is named QuaSpa. Figure 6a illustrates our kernel implementation based on the GPU multi-level cache hierarchy.

### 3.3.1 FUSED WITH PER-TOKEN QUANTIZATION

To minimize the associated memory transfer and computational overhead, we prefetch the corresponding scale factors into registers prior to the MMA. After the kernel completes the FP32 accumulation of the MMA, the accumulated results are scaled by the pre-loaded factors before being written back to global memory. Since the scaling operation is entirely decoupled from the MMA computation, it introduces negligible impact on overall computational efficiency.

### 3.3.2 FUSED WITH TRANSPOSE

Directly fusing the transpose operation introduces additional memory access overhead: the global memory addresses corresponding to the MMA computation results are originally contiguous but become noncontiguous after transposition, thereby forcing what could have been coalesced memory transactions into separate accesses and degrading overall memory efficiency. To integrate the matrix transpose into the sparse matrix multiplication kernel while minimizing the associated overhead, we leverage GPU shared memory to reorganize the process of writing back MMA results to global memory. This write-back procedure can be decomposed into two steps: (1) writing from registers to shared memory and (2) writing from shared memory to global memory. The comparision between our writeback strategy and the naive strategy is shown in Figure 6b. By introducing relatively low-cost memory transfers from registers to shared memory (R2S), we significantly reduce the number of memory transfers from shared memory to global memory (S2G), thereby decreasing memory access time accordingly.

### 3.3.3 SPARSE GROUPGEMM

For Transformer models based on the Mixture-of-Experts (MoE) architecture, GroupGEMM kernels are commonly employed to accelerate computations in the MLP layers. Similarly to GEMM,

Figure 6: (a): Our QuaSpa kernel design. (b) The comparision between our writeback strategy and naive one.

GroupGEMM can also benefit from quantization combined with sparsification for acceleration. We have implemented an extension of QuaSpa to support sparse GroupGEMM with per-token quantization, enabling efficient inference acceleration for MoE-based DiTs.

# 4 EXPERIMENTS

## 4.1 EXPERIMENTS SETUP

Our experiments primarily focus on two types of DiT models: the open-source FLUX.1-dev model and our closed-source model MoE model FLUX-MoE. FLUX.1-dev is a 12B DiT-based model with 19 MMDiT Transformer Blocks (T) and 38 Single Transformer Blocks (S). FLUX-MoE is a 18B DiT model based on the mixture of experts (MoE) architecture.

For FLUX.1-dev, we sparsify the GEMM weights of image modality in the MMDiT Transformer blocks and all GEMM weights in the Single Transformer blocks. For FLUX-MoE, we sparsify the all GroupGEMM weights in MLP layers.

The experiments are performed on 1 GPU server equipped with 8×NVIDIA-L20 GPUs with Ada Lovelace architecture. Each NVIDIA-L20 GPU delivers 239 TFLOPS of dense FP8 compute performance and 478 TFLOPS of sparse FP8 compute performance.

Table 1: GEMM shapes in model inference and efficiency comparison between **FP8 qGEMM** and **FP8 QuaSpa**. The values outside the parentheses represent execution time in milliseconds (ms), and the values inside the parentheses denote Model FLOPs Utilization (MFU).

| Resolution | Part | M | N | K | FP8 qGEMM | FP8 QuaSpa | Speedup |
|---|---|---|---|---|---|---|---|
| **512 × 512** | $T/qkvo\_proj$ | 4096 | 3072 | 3072 | 0.39 (80.1%) | 0.22 (71.0%) | 1.77× |
| | $T/up\_proj$ | 4096 | 12288 | 3072 | 1.41 (92.3%) | 0.86 (75.8%) | 1.64× |
| | $T/down\_proj$ | 4096 | 3072 | 12288 | 1.64 (78.9%) | 0.76 (85.1%) | 2.16× |
| | $S/qkvo\_proj$ | 6144 | 3072 | 3072 | 0.57 (82.2%) | 0.33 (71.0%) | 1.72× |
| | $S/up\_proj$ | 6144 | 12288 | 3072 | 2.10 (93.0%) | 1.27 (77.0%) | 1.65× |
| | $S/down\_proj$ | 6144 | 3072 | 12288 | 2.36 (82.2%) | 1.10 (88.2%) | 2.15× |
| **1024 × 1024** | $T/qkvo\_proj$ | 16384 | 3072 | 3072 | 1.59 (78.9%) | 0.85 (73.5%) | 1.87× |
| | $T/up\_proj$ | 16384 | 12288 | 3072 | 5.72 (91.1%) | 3.48 (74.9%) | 1.64× |
| | $T/down\_proj$ | 16384 | 3072 | 12288 | 6.24 (82.9%) | 2.97 (87.1%) | 2.10× |
| | $S/qkvo\_proj$ | 18432 | 3072 | 3072 | 1.80 (80.8%) | 0.99 (73.5%) | 1.81× |
| | $S/up\_proj$ | 18432 | 12288 | 3072 | 6.50 (90.2%) | 3.91 (75.0%) | 1.66× |
| | $S/down\_proj$ | 18432 | 3072 | 12288 | 7.00 (83.1%) | 3.35 (86.9%) | 2.09× |

Table 2: Single-step DiT inference latency of different strategies at different resolutions.

| Time (ms) | | BF16 sparse | FP8 quantization | FP8 QuaSpa |
|---|---|---|---|---|
| **FLUX.1-dev** | **512 × 512** | 201 (1.33×) | 192 (1.27×) | 151 |
| | **1024 × 1024** | 608 (1.35×) | 561 (1.25×) | 449 |
| **FLUX-MoE** | **2048 × 2048** | 182 (1.24×) | 168 (1.14×) | 147 |
| | **3072 × 3072** | 436 (1.21×) | 403 (1.12×) | 359 |
| | **4096 × 4096** | 934 (1.17×) | 873 (1.09×) | 800 |

## 4.2 KERNEL PERFORMANCE EVALUATION

In this section, we evaluate the efficiency of our QuaSpa. We primarily evaluate the computation time and Model FLOPs Utilization (MFU) of two kernels under several representative GEMM shapes from the FLUX.1-dev model: (1) FP8 GEMM kernel with per-token quantization (**FP8 qGEMM**) (2) FP8 QuaSpa. The experimental results are shown in Table 1. QuaSpa achieves 1.64–2.16× performance improvement over qGEMM while maintaining the MFU above 70%.

## 4.3 END-TO-END IMAGE GENERATION PERFORMANCE EVALUATION

In this section, we evaluate the compute performance and generation quality of our method in end-to-end inference tasks.

### 4.3.1 COMPUTE EFFICIENCY

We compare the efficiency of our hybrid method with FP8 quantization and BF16 sparsification methods. For the FLUX.1-dev model, we evaluated its image generation efficiency at resolutions of 512 × 512 and 1024 × 1024. For FLUX-MoE, we evaluated its image generation efficiency at resolutions of 2048 × 2048, 3072 × 3072 and 4096 × 4096. Table 2 summarizes the experimental results and our method achieves a speedup of 1.09 − 1.35 × compared to existing methods.

### 4.3.2 GENERATION QUALITY

Table 3: Quantitative evaluation of our method. Origin refers to the original model, QuaSpa refers to the model after quantization and sparsification.

| | model | CLIP Score | Pick Score | Image Reward |
|---|---|---|---|---|
| **FLUX.1-dev** | Origin | 33.48 | 22.47 | 0.95 |
| | QuaSpa | 33.97 | 22.40 | 0.92 |
| **FLUX-MoE** | Origin | 34.08 | 23.27 | 1.43 |
| | QuaSpa | 34.35 | 23.19 | 1.44 |

We evaluated the quantitative comparison and qualitative comparison of image generation between our compressed model and the original model.

**Quantitative Comparison.** We evaluated the generation quality metrics including CLIPScore (Hessel et al., 2021), PickScore (Kirstain et al., 2023), and ImageReward (Xu et al., 2023) on MS-COCO (Lin et al., 2014) dataset. The results in Table 3 show that the metrics of our compressed model are close to those of the original model.

**Qualitative Comparison.** We compare the image generation results of the model after our quantization and pruning with those of the original model, as shown in Figure 7. The experimental results demonstrate that our model compression approach has minimal impact on overall generation quality.

## 4.4 ABLATION STUDY OF KERNEL DESIGN

We evaluate the performance benefits of our kernel-level optimizations. As discussed in Section 3.3, our kernel incorporates two key optimizations: (1) fusing the kernels of GEMM workflow into a single

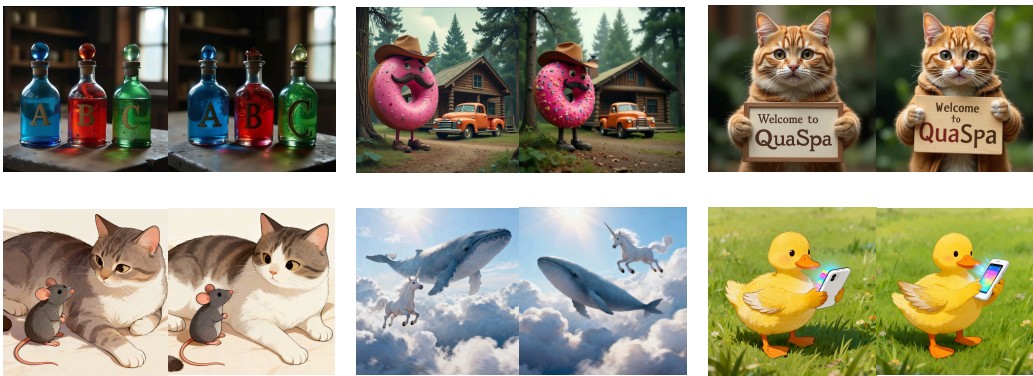

Figure 7: Comparison of generation quality between our sparse quantization model and the original model. In each pair, the left image is generated by the original model, and the right image by the sparsely quantized model, using the same prompt and random seed. The top three pairs show results from the FLUX.1-dev model at 1024 × 1024 resolution, while the bottom three pairs show results from the MoE model at 2048 × 2048 resolution.

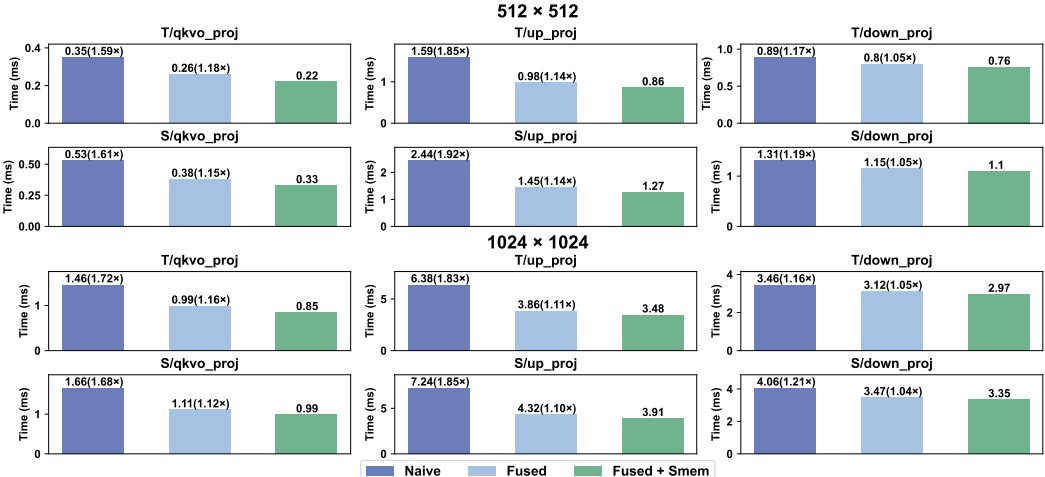

Figure 8: Ablation study of GEMM and transpose kernel. **Naive**: no kernel fusion. **Fused**: kernel fusion. **Fused + Smem**: using shared memory for memory access optimization after kernel fusion (QuaSpa).

kernel. (2) leveraging shared memory to improve the memory access efficiency of the transpose. We conduct corresponding ablation studies to validate the effectiveness of these optimization techniques. The experimental results are shown in the Figure 8.

## 5 CONCLUSION

This paper presents a co-design of algorithm and system that combines model quantization and sparsification to optimize the inference efficiency of DiT models. We obtain sparse models through offline pruning and develop an efficient FP8 quantized sparse matrix multiplication kernel by leveraging kernel fusion and memory access optimizations, fully utilizing the sparse computing capabilities of Tensor Cores for acceleration. The proposed approach achieves better performance compared to existing quantization or pruning methods, while maintaining a negligible impact on overall image generation quality. Our kernel design can be extended to other quantization precisions and to GPU architectures that provide sparse computing capabilities. Furthermore, our method is compatible with other techniques such as model distillation and sparse attention, enabling further improvements in inference efficiency when combined.

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

## A  USAGE OF LARGE LANGUAGE MODELS

We use large language model to polish our writing, including the Introduction and Method. We take full responsibility for the content of the paper, including any text generated or polished by the LLM. We ensure that the LLM-generated text adheres to ethical guidelines and does not contribute to plagiarism or scientific misconduct.

## B  ETHICS STATEMENT

This work adheres to the ICLR Code of Ethics. In this study, no human subjects or animal experimentation was involved. The datasets used in the experiments were sourced according to relevant usage guidelines, ensuring no violation of privacy. We ensure the absence of biases and discriminatory outcomes throughout our research. The study does not include any personally identifiable information, and all experiments were carried out without raising any privacy or security concerns.

## C  REPRODUCIBILITY STATEMENT

We have made every effort to ensure that the results presented in this paper are reproducible. Our method, including kernel design, model compress strategy, and hardware details, is described in detail in the paper. We believe these measures will facilitate the reproducibility of our work and help advance the field.

