# OpenReview forum: "Quantization Meets Sparsification for Faster Image Generation"
_ICLR.cc/2026/Conference — Submitted to ICLR 2026_

### Official Review · Reviewer_CkCN · 2025-10-24

**Soundness:** 2
**Presentation:** 2
**Contribution:** 2
**Rating:** 4
**Confidence:** 3

**Summary:**

This paper proposed QuaSpa, a sparsification and quantization kernel designed for efficient image generation. It develop a novel and efficient matrix multiplication kernel that leverages low-bit and sparse computing capabilities of Tensor Cores. Experiments on FLUX show that QuaSpa achieves a speedup of 1.64-2.16 times, delivering an efficiency improvement of 1.09-1.35 times to the end-to-end workflow.

**Strengths:**

1.The topic of Quaspa is meaningful, and combining sparsity with quantization can further promote the development of efficient visual generation.

2.QuaSpa has practical hardware implementation that can achieve acceleration effects in real-world inference scenarios.

3.QuaSpa can basically guarantee the performance of the model and bring actual acceleration.

**Weaknesses:**

1.What is the actual quantization setting? For example, is it symmetric quantization, and does the quantization granularity of weights support per-channel or per-group?

2.The paper only implement FP8 quantization, whether it can support more settings such as INT8, INT4, and FP4.

3.The sparsity ratio supported by the paper is relatively fixed at only 2:4. Is there a corresponding theoretical support method for other sparsity settings?

4.The specific settings for FLUX evaluation in the paper are unknown, such as the dataset and sample number.

**Questions:**

Please see above weaknesses.

---

### Official Review · Reviewer_hU37 · 2025-10-31

**Soundness:** 2
**Presentation:** 3
**Contribution:** 2
**Rating:** 2
**Confidence:** 4

**Summary:**

This paper introduces a system for accelerating image generation in large diffusion transformer models (specifically DiT architectures) by combining offline pruning-based sparsification with per-token quantization. The framework includes both a workflow for rapidly deploying sparse-quantized model variants and the design of QuaSpa, an efficient fused kernel for quantized sparse GEMM utilizing GPU Tensor Core sparse capabilities. Extensive experiments on both open-source (FLUX.1-dev, 12B) and industrial MoE (18B) models demonstrate notable performance gains, with speedups of 1.64–2.16× in kernel time and around 1.09–1.35× in end-to-end inference, all with minimal loss in output image quality.

**Strengths:**

- Engineering addresses real hardware bottlenecks. The paper explains why sparsity must live on matrix A for Tensor Cores, motivates swapping A/B, and then fuses quantization scaling and the required transpose into the sparse MMA kernel (“QuaSpa”), with a clear cache-hierarchy dataflow (Sec. 3.3; Fig. 5–6).
- Performance gains are concrete (kernel + single-step). QuaSpa delivers 1.64–2.16× kernel-level speedups over FP8 per-token qGEMM with MFU ≥ 70% (Table 1) and 1.09–1.35× end-to-end single-step latency reductions across models/resolutions (Table 2; Sec. 4.2–4.3.1).
- Exposition is clear and well structured. The workflow, where the extra scaling/transpose costs arise, and how the fusion resolves them are described coherently and aligned with the figures (Sec. 3.3; Fig. 5–6).

**Weaknesses:**

- Only single-step latency is reported; full sampling throughput is unknown. Table 2 focuses on single-step numbers. There are no results for total wall-clock/throughput (images/s) under standard multi-step samplers (e.g., 50 steps) at fixed batch sizes, making real deployment gains hard to judge (Sec. 4.3.1).
- Detail fidelity is not quantified. Reported metrics (CLIPScore, PickScore, ImageReward) emphasize semantic alignment and preference proxies; there is no measurement of fine-grained structure/edges or text readability. Fig. 7 provides qualitative comparisons but no quantitative analysis of detail changes (Sec. 4.3.2).
- Baseline coverage is narrow. End-to-end results mainly contrast BF16 sparse, FP8 quantized, and FP8 QuaSpa. Stronger engineering baselines (e.g., cuBLASLt/CUTLASS 2:4 FP8 or a dense W8A8 recipe like SmoothQuant) are absent (Sec. 4.3.1).
- Generalization beyond 2:4 sparsity is untested. The method and rationale target NVIDIA 2:4 Tensor Core sparsity; there is no evidence for other (n:m) patterns or how performance/quality would change (Sec. 3.3; Fig. 5).

**Questions:**

- End-to-end performance: Under standard multi-step sampling (e.g., 50/75 steps) with fixed sampler/CFG, what are total wall-clock times and throughput (images/s) at 512²/1024² for FLUX.1-dev and 2048²/3072² for the MoE model? Please specify batch sizes and report mean ± std over ≥ 3 seeds (Table 2 currently reports only single-step).
- Compression sensitivity: How do CLIPScore/PickScore/ImageReward vary as sparsity and quantization granularity (per-token vs per-block) change? Can you include tolerance/ non-inferiority curves and a short failure-case analysis (Sec. 4.3.2)?
- (n:m) formats: Can QuaSpa support sparsity patterns beyond 2:4? If yes, what speed/overhead changes should we expect; if not, where are the limiting factors (instructions, data layout, or fusion points) (Sec. 3.3; Fig. 5)?
- Reproducibility: Will you open-source the code to reproduce Table 1–2 (kernel & single-step) and Table 3 (quality) results (Sec. 4.2–4.3)?

---

### Official Review · Reviewer_LEXr · 2025-10-31

**Soundness:** 2
**Presentation:** 2
**Contribution:** 1
**Rating:** 4
**Confidence:** 3

**Summary:**

The paper proposes QuaSpa, a new sparse GEMM kernel that combines 2:4 structured sparsity with FP8 per-token quantization for accelerated inference of Diffusion Transformer Models (DiTs). QuaSpa is built on top of NVIDIA’s sparse Tensor Core, and leverages kernel fusion and memory access optimizations for more efficient computations. Experiments are performed on two DiT models, and results show an end-to-end speedup of 1.09-1.35×, while maintaining image quality.

**Strengths:**

* This work addresses an important problem: accelerating inference of diffusion models is extremely relevant, as they are slow and computationally expensive. Therefore, acceleration in this area has a significant impact.
* There is a solid system contribution. The kernel-level optimization (fused strategy and shared memory optimization are technically sound and valuable.
* The figures are detailed and help in the understanding of the paper details.
* The experiments show actual runtime gains over the chosen baselines.

**Weaknesses:**

* The paper is missing strong baselines. The comparisons are limited to a dense FP8 baseline (“qGEMM”) (4.2) and ablations where either quantization or sparsification is applied in isolation (4.3). The authors should include comparisons against existing sparse kernels, such as NVIDIA’s Sparse Tensor Core implementation [1] with quantization. Since the goal is efficient generation, other baselines could be included such as Q&C [2], which does not use sparsity but has a similar goal.
* Limited generability: Although the title mentions accelerating image generation, only DiTs are tested. Since QuaSpa is architecture agnostic, testing another model class would strengthen the paper.
* Modest novelty: The work is an engineering integration of existing ideas, both already supported in NVIDIA hardware and libraries. The main novelty comes from the kernel-level fusion, and without strong baselines it is difficult to assess the true benefit of the proposed design.
* Several important details are missing:
  - The authors don’t expand on the type of pruning applied, only mentioning that they follow “existing offline model pruning methodologies“.
  - The definitions of qGEMM, BF16 sparse, and FP8 quantization (Table 1 and Table 2) are not clearly explained, making them ambiguous.
  - The generation quality is evaluated on MS-COCO. Does this mean that the authors used the captions? How many images were created?


[1] Asit Mishra, Jorge Albericio Latorre, Jeff Pool, Darko Stosic, Dusan Stosic, Ganesh Venkatesh, Chong Yu, Paulius Micikeviciu (2021) Accelerating Sparse Deep Neural Networks.

**Questions:**

1. How does QuaSpa compare quantitatively to NVIDIA’s Sparse Tensor Core implementation with quantization?
2. Can you include FID to assess Generation Quality in 4.3.2?
3. How does QuaSpa compare with Q&C [2]? Could they run jointly?
4. How does QuaSpa perform on other image generation models?


[2] Xin Ding and Xin Li and Haotong Qin and Zhibo Chen (2025) Q&C: When Quantization Meets Cache in Efficient Image Generation

---

### Official Review · Reviewer_n6bo · 2025-10-31

**Soundness:** 3
**Presentation:** 3
**Contribution:** 2
**Rating:** 6
**Confidence:** 3

**Summary:**

This paper introduces a unified acceleration framework that combines quantization and sparsification to speed up image generation in large Diffusion Transformer (DiT) models. Unlike prior works that apply these compression techniques separately, the authors integrate them into a single system that leverages both low-bit precision and structured sparsity. The method, implemented through a custom GPU kernel called QuaSpa, supports per-token FP8 quantization and exploits the 2:4 structured sparsity pattern available on NVIDIA Tensor Cores. By jointly applying offline pruning and dynamic quantization, and fusing quantization, sparsification, and transpose operations within the kernel, the approach minimizes memory access overhead and maximizes computational efficiency.

Experiments on the FLUX.1-dev (12B) and FLUX-MoE (18B) models demonstrate that the proposed method achieves 1.64–2.16× kernel-level speedups and 1.09–1.35× end-to-end acceleration, with almost no loss in image generation quality as measured by CLIPScore, PickScore, and ImageReward. The resulting images are visually indistinguishable from those of the original models. Overall, the paper makes a strong contribution by showing that quantization and sparsification can be effectively co-designed at both the algorithmic and system levels, yielding a scalable and hardware-efficient solution for accelerating large diffusion-based generative models.

**Strengths:**

The paper’s main strengths lie in its innovative integration of quantization and sparsification and its strong hardware–software co-design. By jointly applying per-token FP8 quantization and 2:4 structured sparsity, the proposed method effectively reduces both memory bandwidth and computation costs—two of the main bottlenecks in large-scale diffusion models. The custom QuaSpa kernel fuses quantization, sparsity, and transpose operations into a single GPU kernel, fully leveraging NVIDIA Tensor Cores while minimizing memory access overhead. This co-optimization between algorithmic design and hardware execution demonstrates a deep understanding of both model compression and GPU architecture.

Another strength is the scalability and practical validation of the approach. The method is tested on large diffusion models—FLUX.1-dev (12B) and FLUX-MoE (18B)—showing up to 2× kernel-level speedups and 1.35× end-to-end acceleration with negligible impact on generation quality. Quantitative and qualitative results confirm that image fidelity remains nearly identical to the uncompressed models. Overall, the work is technically solid, empirically convincing, and highly relevant, presenting a balanced combination of compression theory, system engineering, and real-world generative performance.

**Weaknesses:**

The main weaknesses of the paper lie in its limited generality and evaluation scope. The proposed approach is closely tied to Diffusion Transformer architectures and the 2:4 structured sparsity pattern supported by NVIDIA Tensor Cores, which restricts its portability to other generative models or hardware platforms. This fixed sparsity structure, while efficient for acceleration, may not yield optimal compression or flexibility across different layers or architectures. Moreover, the experimental validation is confined to two large models (FLUX.1-dev and FLUX-MoE) on a single GPU setup, leaving open questions about robustness, scalability, and applicability to broader diffusion or transformer variants.

Another weakness is the paper’s heavy emphasis on engineering optimization over theoretical analysis. While the QuaSpa kernel design is technically sophisticated, the paper provides little insight into the effects of quantization noise, sparsity-induced degradation, or error propagation in diffusion dynamics. Implementing the method also requires considerable GPU programming expertise, which may limit practical adoption. Overall, the work is strong on systems innovation but less comprehensive in theoretical depth, generalization, and experimental diversity.

**Questions:**

1. Generalization to other architectures: Have you evaluated whether the proposed quantization–sparsification framework can be extended to U-Net–based diffusion models or text-conditioned transformers (e.g., Stable Diffusion, SDXL)? If not, what technical barriers prevent this generalization?

2. Sensitivity analysis: How sensitive is the performance and image quality to different quantization precisions (e.g., FP8 vs. INT4) or sparsity levels beyond the fixed 2:4 pattern? Could relaxing the sparsity constraint yield better trade-offs between speed and accuracy?

3. Compatibility with other accelerations: Could the QuaSpa kernel integrate with sparse attention mechanisms or Mixture-of-Experts routing sparsity to achieve further end-to-end acceleration?

4. Cross-hardware portability: Given that the current method relies heavily on NVIDIA Tensor Cores, how might this framework adapt to other GPU or AI accelerator architectures (e.g., AMD ROCm, TPUs, or custom NPUs)?

---

### Meta-Review · Area_Chair_us3N · 2026-01-06

**Summary:**

Reviewers view the paper as a good systems contribution, combining quantization and structured sparsity for accelerating Diffusion Transformer inference via a fused kernel. Strengths include strong system-algorithm co-design and clear kernel-level. Main concerns are limited novelty beyond engineering integration, weak or narrow baselines, restricted generality (neural network architecture and hardware), and evaluation focused on single-step latency rather than full sampling throughput.

**Reviewer Concerns:**

There was no author rebuttal. As a result, reviewer concerns regarding baselines, generality, and evaluation scope remain unaddressed.

**Reviewer Scores:**

Given the absence of a rebuttal or discussion, reviewer scores were unlikely to change, and the majority of them are inclined towards rejection.

---

### Decision · Program_Chairs · 2026-01-26

Reject